# Full genome characterization of 12 citrus tatter leaf virus isolates for the development of a detection assay

Shih-hua Tan[1]ʘ, Fatima Osman[2]ʘ, Sohrab Bodaghi[1], Tyler Dang[1], Greg Greer[1], Amy Huang[1], Sarah Hammado[1], Shurooq Abu-Hajar[1], Roya Campos[1], Georgios Vidalakis[1]*

1 Department of Microbiology and Plant Pathology, University of California, Riverside, California, United States of America, 2 Department of Plant Pathology, University of California, Davis, California, United States of America

ʘ These authors contributed equally to this work.
* georgios.vidalakis@ucr.edu

**Data Availability Statement:** All sequence files characterized in this study are available from the NCBI GenBank database (accession numbers: MH108975 - MH108986). All other relevant data

## Abstract

Citrus tatter leaf virus (CTLV) threatens citrus production worldwide because it induces bud-union crease on the commercially important Citrange (*Poncirus trifoliata × Citrus sinensis*) rootstocks. However, little is known about its genomic diversity and how such diversity may influence virus detection. In this study, full-length genome sequences of 12 CTLV isolates from different geographical areas, intercepted and maintained for the past 60 years at the Citrus Clonal Protection Program (CCPP), University of California, Riverside, were characterized using next generation sequencing. Genome structure and sequence for all CTLV isolates were similar to *Apple stem grooving virus* (ASGV), the type species of *Capillovirus* genus of the *Betaflexiviridae* family. Phylogenetic analysis highlighted CTLV's point of origin in Asia, the virus spillover to different plant species and the bottleneck event of its introduction in the United States of America (USA). A reverse transcription quantitative polymerase chain reaction assay was designed at the most conserved genome area between the coat protein and the 3'-untranslated region (UTR), as identified by the full genome analysis. The assay was validated with different parameters (e.g. specificity, sensitivity, transferability and robustness) using multiple CTLV isolates from various citrus growing regions and it was compared with other published assays. This study proposes that in the era of powerful affordable sequencing platforms the presented approach of systematic full-genome sequence analysis of multiple virus isolates, and not only a small genome area of a small number of isolates, becomes a guideline for the design and validation of molecular virus detection assays, especially for use in high value germplasm programs.

## Introduction

Citrus tatter leaf virus (CTLV), a *Capillovirus* belonging to the family *Betaflexiviridae*, is considered to be a strain of *Apple stem grooving virus* (ASGV) [1, 2]. CTLV is readily transmitted

are within the manuscript and its supporting information files.

**Funding:** This research was funded by the Citrus Research Board (CRB) project "Citrus tatter leaf – Citrange stunt; The Hidden Dragon" (5300-167) awarded to Vidalakis G. Additional support was provided by the United States Department of Agriculture (USDA) National Institute of Food and Agriculture (NIFA), Hatch (project 233744) and the National Clean Plant Network (NCPN) which operates under the auspices of USDA Animal and Plant Health Inspection Service (APHIS) (12-8100-1544-CA; 14-, 15-, 16- 8130-0419-CA; AP17PPQS&T00C118; AP18PPQS&T00C107). None of the authors received salary or other funding from commercial companies. The funders had no role in study design, data collection and analysis, decision to publish, or preparation of the manuscript. URLs to sponsors' websites are listed below. 1.CRB: https://www.citrusresearch.org. 2. USDA NIFA: https://nifa.usda.gov. 3.NCPN: http://nationalcleanplantnetwork.org.

**Competing interests:** The authors have declared that no competing interests exist.

mechanically and no natural vectors have been yet identified [2]. CTLV was first discovered in Chico, California, USA [3, 4] in latent infected Meyer lemon trees (*Citrus Limon* (L.) Burm.f. hyb.), a cultivar imported around 1908 from Asia into the country. CTLV is endemic to China [5, 6] and it has been found in Taiwan [6–8], Japan [9–13], Australia [14, 15], South Africa [16] and in the USA; in California [3], Florida [2, 4, 17] and Texas [18, 19].

Although CTLV was first discovered in citrus, it has been reported to infect a wide range of herbaceous hosts, many of which remain symptomless [13]. Most CTLV infected commercial citrus varieties also remain asymptomatic except when CTLV infected budwood is propagated onto trifoliate orange (*P. trifoliata* (L.) Raf.) or trifoliate hybrid citrange (*P. trifoliata* × *C. sinensis*) rootstocks [2, 20]. The resulting citrus trees are stunted, display chlorotic leaves, and show bud union incompatibility, leading to the ultimate decline of the tree [10, 21]. This poses a serious problem because trifoliate and trifoliate hybrid rootstocks are widely used in all citrus producing areas of the world for their tolerance to citrus tristeza virus and *Phytophthora* species in addition to many other desirable horticultural characteristics (e.g. freeze tolerance, good yield and fruit quality) [22–24].

The numerous asymptomatic citrus and non-citrus hosts in combination with the destructive potential of the virus for trees propagated on commercially important rootstocks make CTLV a serious threat to the citrus industry [17, 20, 21, 25]. Reliable pathogen detection assays for the production, maintenance, and distribution of pathogen-tested propagative materials by citrus germplasm and certification programs are the basis for any successful mitigation effort against viral threats, including CTLV [26–31]. Bioindicators for indexing of CTLV such as *Citrus excelsa*, and Rusk citrange, displaying symptoms of deformed young leaves under controlled greenhouse conditions, provide a reliable diagnostic tool [6]. ASGV antiserum was used both in enzyme-linked immunosorbent assay and in immunocapture RT-PCR for CTLV detection [32]. A series of conventional reverse-transcription polymerase chain reaction (RT-PCR) based methods had been developed for CTLV including two-step multiplex assays [33, 34] and a one-step RT-PCR assay with a semi-nested variation [28]. More recently, reverse transcription quantitative PCR (RT-qPCR) assays were developed for CTLV detection using SYBR® Green [35] and florescent probe platforms [25].

At the time that Liu et al. (2011) published their assay there were only four full-genome CTLV sequences deposited in the GenBank [35]. Cowell et al. (2017) reported that the RT-qPCR assay was designed based on seven full-genome sequences available at the time in the GenBank [25]. Today, a total of 12 full-genome sequences are available in the GenBank [2, 36, 37]. Due to the limited number of CTLV full-genome sequences, very little is known about the phylogenetic relationship and genomic diversity of virus and how such diversity may influence its detection. Next generation sequencing (NGS) technologies combined with bioinformatics have proven to be powerful tools for the assembly of full-genome virus sequences [38–40] and the guidelines for the design and validation of real-time qPCR methods are well established [41, 42]. The purpose of this study was to characterize and further develop a robust CTLV RT-qPCR detection assay based on the systematic analysis of newly generated full-length genome data from multiple virus isolates maintained for the past 60 years at the CCPP.

## Materials and methods

### Virus isolates and RNA extraction for full-length genome sequencing

Twelve CTLV isolates from various citrus varieties introductions, originating from different geographical locations, were intercepted and maintained *in planta* under quarantine at the CCPP disease collection between 1958 and 2014 (Table 1). Sweet orange (*C. sinensis* (L.) Osbeck) seedlings were graft-inoculated with the different CTLV isolates and total RNA was

**Table 1. Isolates of citrus tatter leaf virus used in this study.**

| Sample | Citrus Host | Citrus Host Scientific Name | Geographic Origin | Isolation Year | Biological Indexing[1] | Genome Size (nt) | GenBank Accession No. |
|---|---|---|---|---|---|---|---|
| IPPN122 | Sa Tou Satsuma | *C. unshiu* (Macf.) Marc. | China | 1992 | NA | 6497 | MH108986 |
| TL100 | Meyer Lemon | *C. limon* (L.) Burm.f. hyb. | TX, USA | 1958 | 3 | 6495 | MH108975 |
| TL101 | Meyer Lemon | *C. limon* (L.) Burm.f. hyb. | CA, USA | 1969 | 3 | 6494 | MH108976 |
| TL102 | Meyer Lemon | *C. limon* (L.) Burm.f. hyb. | CA, USA | 1958 | 3 | 6495 | MH108977 |
| TL103 | Hirado Buntan Pummelo | *C. grandis* (L.) Osb. | Japan | 1983 | NA | 6495 | MH108978 |
| TL104 | Kobeni Mikan Tangor | *C. reticulata* x *C. sinensis* | China | 1987 | NA | 6495 | MH108979 |
| TL110 | Little Sweetie Satsuma | *C. unshiu* (Macf.) Marc. | CA, USA | 1989 | NA | 6495 | MH108980 |
| TL111 | Meyer Lemon | *C. limon* (L.) Burm.f. hyb. | FL, USA | 1964 | NA | 6495 | MH108981 |
| TL112 | Citron | *C. medica* L. | China | 2014 | NA | 6496 | MH108982 |
| TL113 | Citron | *C. medica* L. | China | 2014 | NA | 6496 | MH108983 |
| TL114 | Citron | *C. medica* L. | China | 2014 | NA | 6496 | MH108984 |
| TL115 | Dekopan Tangor | *C. reticulata* x *C. sinensis* | Japan | 2007 | NA | 6495 | MH108985 |

[1] The biological indexing was performed on *Citrus exclesa* and Rusk citrange. Symptom scores from 0 (no symptom) to 5 (severe symptoms).

extracted from phloem-rich bark tissues of the last matured vegetative flush (i.e. one-year-old budwood) using TRIzol® reagent (Invitrogen, Carlsbad, California, USA) per manufacturer's instructions. The purity and concentration of the RNA were tested using a Nanodrop spectrophotometer and Agilent 2100 Bioanalyzer per manufacturer's instructions.

## NGS library preparation and bioinformatics

CTLV RNA libraries were constructed using 4μg of total RNA with TruSeq Stranded mRNA Library Prep Kit (Illumina, San Diego, California, USA) per manufacturer's instructions. The RNA libraries were sequenced on an Illumina HiSeq 2500 instrument with high-output mode and single-end 50 or 100 base pairs (bp) at SeqMatic LLC (Fremont, California, USA). All sequencing data was generated by SeqMatic using an Illumina Genome Analyzer IIx and filtered through the default parameters of the Illumina QC pipeline and demultiplexed. The files were uploaded onto the VirFind bioinformatics server and mapped to the reference genome by Bowtie 2, followed by outputting mapped and unmapped contig sequences [43]. Unmapped sequences were *de novo* assembled by Trinity [43]. Assembled contigs were analyzed through BLASTn with an E-value cutoff of $10^{-2}$ against all virus sequences in GenBank and generated outputs of reads and report for virus sequences.

## Rapid amplification of cDNA ends of viral RNA

The 5' and 3' end sequences were obtained via rapid amplification of cDNA ends (RACEs). The 5' end sequence of each CTLV isolate was confirmed using FirstChoice® RLM-RACE Kit (Thermo Fisher Scientific, Carlsbad, California, USA). As per manufacturer's instructions, first-strand cDNA was synthesized and followed by nested PCR with the primer sets listed in S1 Table. To confirm the 3' end sequence of each CTLV isolate, first-strand cDNA was synthesized using SuperScript® II transcriptase (Thermo Fisher Scientific, Carlsbad, California, USA) with oligo dT 16mer and then performed PCR using Platinum® Taq DNA Polymerase High Fidelity Kit (Thermo Fisher Scientific, Carlsbad, California, USA) with the oligo dT 16mer and a CTLV gene specific primer (S1 Table). The PCR product that contained either the 5' or 3' end was ligated into pGEM®-T Easy Vector Systems (Promega, Madison, Wisconsin, USA) per manufacturer's instructions and sequenced using both T7 (5'-TAATACGACTC

ACTATAGGG-3') and SP6 (5'-ATTTAGGTGACACTATAG-3') primers. Together with the contigs containing CTLV sequences from NGS, the sequence data were then analyzed and assembled as consensus full-length genome, using Vector NTI Advance™11 software (Thermo Fisher Scientific, Carlsbad, California, USA).

## Phylogenetic and genomic identity analysis of full-length virus sequences

Phylogenetic analysis was performed using the Molecular Evolutionary Genetics Analysis tool (MEGA version 7.0.21) [44]. ClustalW was used to align the 12 newly generated CTLV full-length cDNA sequences with the capilloviruses: CTLV, ASGV, pear black necrotic leaf spot virus (PBNLSV; a strain of ASGV), and cherry virus A (CVA) for which full genome sequences were available in GenBank (Table 2). Phylogenetic topologies were reconstructed using three different methods: neighbor-joining, maximum likelihood and minimum evolution and tested with 1,000 bootstrap replicates. All phylogenetic methods gave similar results and the neighbor-joining tree was presented in this study. Nucleotide (nt) percentage of sequence identities were calculated for CTLV complete or partial genomes using the pairwise sequence identity and similarity in a web-based analyzing program (http://imed.med.ucm.es/Tools/sias.html).

## Citrus sample processing and RNA extraction for RT-qPCR detection of CTLV

To account for the possible uneven distribution of the virus within a plant, budwood samples from four to six different branches around the tree canopy were randomly collected and combined in a single sample. Samples from the citrus trees' phloem-rich bark of matured budwood (approximately 12 to 18 months old) were collected and processed by freeze-drying and grinding as described by Osman et al. 2017 [45]. Total RNA was extracted from the ground sample using MagMAX™ Express-96 (Thermo Fisher Scientific, Carlsbad, California, USA) along with a modified 5X MagMax™-96 Viral RNA Isolation Kit optimized for citrus tissues [45]. Total RNA was eluted in 100 μl elution buffer and used as template for RT-qPCR.

## RT-qPCR assay design

For the specific detection of CTLV in citrus tissues, an RT-qPCR assay was designed based on sequence conservation alignment of a total 28 full genome sequences: 23 sequences of CTLV, (12 generated in this study and 11 from the GenBank) and five GenBank sequences of ASGV isolated from citrus and kumquat, a citrus relative (S1 Fig). Primers and probe were designed using the Primer Express™ software (Thermo Fisher Scientific, Carlsbad, California, USA) and following the guidelines for designing RT-qPCR assays a 58°C optimum melting temperature for primers and a 10°C increase for qPCR probes was used to prevent the formation of primer dimers (Table 3). The fluorophore used for the CTLV probe was 6-carboxyfluorescein FAM and the 3' quencher was Black Hole Quencher (BHQ). The homology of the primers and qPCR probe was confirmed by a BLAST search against the GenBank database.

The RT-qPCR reaction (12 μl total volume) was performed using the AgPath-ID™ One-Step RT-PCR Kit (Thermo Fisher Scientific, Carlsbad, California, USA) with 2.65 μL water, 6.25 μL 2X RT buffer, 0.6 μL primer probe mix (417 nM for primers and 83 nM for probe as final concentrations), 0.5 μL 25X RT mix and 2 μL of RNA for each reaction. The cycling conditions were 45°C for 10 minutes, 95°C for 10 minutes during the first cycle, followed by 40 cycles of 95°C for 15 seconds and 60°C for 45 seconds. Samples were analyzed using Applied Biosystems™ 7900HT Fast Real-Time PCR System and Applied Biosystems™ QuantStudio 12K Flex Real-Time PCR System (Thermo Fisher Scientific, Carlsbad, California, USA). Fluorescent signals were collected during the amplification cycle and the quantitative cycle (Cq) was

**Table 2. Full-length nucleotide sequences of citrus tatter leaf virus isolates and capilloviruses used in phylogenetic and sequence identity analysis.**

| Isolate | Host | Host Scientific Name | Geographic Origin | Isolation Year | GenBank Accession Number | GenBank Deposit Year | Cluster | Clade |
|---|---|---|---|---|---|---|---|---|
| AGSV-YTG | Apple | *Malus domestica* | China | 2012 | KJ579253 | 2014 | I | A |
| ASGV-HH | Pear | *Pyrus pyrifolia* cv. 'Huanghua' | China | 2009 | JN701424 | 2012 | | |
| ASGV-CHN | Apple | *M. domestica* | China | 2011 | JQ308181 | 2013 | | |
| ASG-241KP | Apple | *M. domestica* | Japan | 1992 | D14995 | 2008 | | |
| ASGV-P-209 | Apple | *M. domestica* | Japan | 1993 | NC001749 | 2018 | | |
| ASGV-Nagami | Kumquat | *Fortunella margarita* (Lour.) Swing. | Japan | 2016 | LC184612 | 2017 | | |
| CTLV-ASGV-2-HJY | Citrus- Huang Jin Mi You | *C. maxima* (Burm.) Merrill | China | 2016 | MH144343 | 2018 | | |
| CTLV-MTH | Citrus- Ponkan Mandarin | *C. reticulata* Blanco | China | 2013 | KC588948 | 2013 | | |
| CTLV-IPPN122 | Citrus- Sa Tou Satsuma | *C. unshiu* (Macf.) Marc. | China | 1992 | MH108986 | 2018 | | |
| CTLV-L | Lily | *Lilium longiflorum* | Japan | 1993 | D16681 | 2008 | II | |
| ASGV-Li-23 | Apple | *M. domestica* | Japan | 1997 | AB004063 | 2000 | | |
| ASGV-FKSS2 | Citrus | *C. junos* Sieb. ex Tanaka | Japan | 2014 | LC143387 | 2016 | | |
| ASGV-N297 | Citrus- Satsuma | *C. unshiu* (Macf.) Marc. | Japan | 1987 | LC184610 | 2017 | | |
| ASGV-AC | Apple | *M. domestica* | Germany | 2009 | JX080201 | 2012 | III | B |
| ASGVp12 | Apple | *M. domestica* cv. Red Chief | India | 2011 | HE978837 | 2015 | | |
| ASGV-Ac | Actinidia | *Actinidia* sp. | China | 2015 | KX988001 | 2017 | | |
| ASGV-Matsuco | Citrus | *C. tamurana* | Japan | 2014 | LC084659 | 2015 | | |
| CTLV-Ponkan8 | Citrus- Ponkan Mandarin | *C. reticulata* Blanco | Taiwan | 2012 | KY706358 | 2018 | | |
| CTLV-Pk | Citrus- Ponkan Mandarin | *C. reticulata* Blanco | Taiwan | 2012 | JX416228 | 2012 | | |
| CTLV-TL113 | Citrus- Citron | *C. medica* L. | China | 2014 | MH108983 | 2018 | | |
| CTLV-TL114 | Citrus- Citron | *C. medica* L. | China | 2014 | MH108984 | 2018 | | |
| CTLV-TL112 | Citrus- Citron | *C. medica* L. | China | 2014 | MH108982 | 2018 | | |
| CTLV-LCd-NA-1 | Citrus- Sweet Orange | *C. sinensis* L. Osb. | Taiwan | 2004 | FJ355920 | 2008 | | |
| CTLV-HJY | Citrus- Huang Jin Mi You | *C. maxima* (Burm.) Merrill | China | 2016 | MH144341 | 2018 | | |
| CTLV-Kumquat1 | Kumquat | *F. margarita* (Lour.) Swing. | Taiwan | 2004 | AY646511 | 2004 | | |
| CTLV-Shatang Orange | Citrus- Shatang Mandarin | *C. reticulata* Blanco | China | 2011 | JQ765412 | 2012 | | |
| CTLV-XHC | Citrus- Sweet Orange | *C. sinensis* L. Osb. | China | 2013 | KC588947 | 2013 | | |
| CTLV-ML | Citrus- Meyer Lemon | *C. limon* (L.) Burm.f. hyb. | FL, USA | 2008 | EU553489 | 2010 | IV | |
| CTLV-TL111 | Citrus- Meyer Lemon | *C. limon* (L.) Burm.f. hyb. | FL, USA | 1964 | MH108981 | 2018 | | |
| CTLV-TL110 | Citrus- Little Sweetie Satsuma | *C. unshiu* (Macf.) Marc. | CA, USA | 1989 | MH108980 | 2018 | | |
| CTLV-TL103 | Citrus- Hirado Buntan Pummelo | *C. maxima* (Burm.) Merrill | Japan | 1983 | MH108978 | 2018 | | |
| CTLV-TL101 | Citrus- Meyer Lemon | *C. limon* (L.) Burm.f. hyb. | CA, USA | 1969 | MH108976 | 2018 | | |
| CTLV-TL100 | Citrus- Meyer Lemon | *C. limon* (L.) Burm.f. hyb. | TX, USA | 1958 | MH108975 | 2018 | | |
| CTLV-TL102 | Citrus- Meyer Lemon | *C. limon* (L.) Burm.f. hyb. | CA, USA | 1958 | MH108977 | 2018 | | |
| CTLV-TL104 | Citrus- Kobeni Mikan Tangor | *C. reticulata* x *C. sinensis* | China | 1987 | MH108979 | 2018 | | |
| CTLV-TL115 | Citrus- Dekopon Tangor | *C. reticulata* x *C. sinensis* | Japan | 2007 | MH108985 | 2018 | | |
| ASGV-Kiyomi | Citrus | *C. unshiu* x *C. sinensis* | Japan | 2016 | LC184611 | 2017 | | |
| CTLV-ASGV-1-HJY | Citrus- Huang Jin Mi You | *C. maxima* (Burm.) Merrill | China | 2016 | MH144342 | 2018 | | |
| PBNLSV | Pear | *P. pyrifolia* | S. Korea | 2004 | AY596172 | 2004 | Outgroup | |
| ASGV-kfp | Pear | *P. pyrifolia* | China | 2014 | KR106996 | 2015 | | |
| AGSV-HT | Apple | *M.* spp. Crabapple | China | 2015 | KU947036 | 2017 | | |
| CVA | Cherry | *Prunus avium* L. cv. Sam | Germany | 1994 | NC003689 | 2018 | | |

Abbreviations: CTLV: citrus tatter leaf virus; ASGV: apple stem grooving virus; PBNLSV: pear black necrotic leaf spot virus; CVA: cherry virus A; S. Korea: South Korea

**Table 3. Oligonucleotide primers and probe of citrus tatter leaf virus detection assay designed in this study.**

| Primers/probes* | Sequence 5'- 3' | Nucleotide Position[1] | Amplicon size (bp) |
|---|---|---|---|
| CTLV 6315 F1 | CGAGGCAGGTTCGGAGAGTA | 6315–6334 | 82 |
| CTLV 6316 F2 | GAGGCGGGTTCGGAGAGTA | 6316–6334 | |
| CTLV 6315 F3 | TGAGGCAGGTTCGGAGAGTAA | 6315–6335 | |
| CTLV R | CCTGCAAGACCGCGACC | 6380–6396 | |
| CTLV 6338 P FAM | TGGAACTGGAGGGTTAG | 6338–6354 | |

[1]Nucleotide position is based on reference genome of citrus tatter leaf virus isolate TL100 (NCBI GenBank Accession No. MH108975).

*F: forward primer. R: Reverse primer. P: qPCR probe.

calculated and exported with a threshold of 0.2 and a baseline of 3–15 for the targets of interest. The Cq was calculated by the qPCR machine using an algorithm with a set range of cycles at which the first detectable significant increase in fluorescence occurs. RNA and reaction integrity were assessed using the qPCR assay targeting cytochrome oxidase (COX) gene in the citrus genome as the internal control [27].

## RT-qPCR assay validation

The newly designed CTLV RT-qPCR assay was validated using applicable parameters proposed in the "Guidelines for validation of qualitative real-time PCR methods" [41]. Applicability, practicability and transferability were evaluated by deploying the assay at two different laboratories, University of California (UC) Riverside- CCPP and UC Davis- Real-Time PCR Research & Diagnostic Core Facility. The robustness of the assay was evaluated with deviation in annealing temperatures (±2 ºC), reaction volumes (±2 μL), and different qPCR instruments (CFX96 Real-Time PCR Detection System, Bio-Rad, Hercules, CA), and master mixes (iTaq™ Universal Probes One-Step Kit, Bio-Rad, Hercules, CA) to optimize the assay.

The specificity of the assay was evaluated both *in silico* and experimentally, using a variety of citrus samples with known CTLV infection status from broad geographical origins and isolation times. All virus isolates exotic to California were received as nucleic acids under the auspices of the United States Department of Agriculture (USDA) Animal and Plant Health Inspection Service (APHIS) Plant Protection and Quarantine (PPQ) permits P526P-18-04608 and P526P-18-04609. Cross-reactivity was assessed using RNA of different non-inoculated citrus species and varieties and RNA from citrus inoculated with other non-targeted graft-transmissible pathogens of citrus.

The sensitivity (absolute limit of detection, $LOD_6$) and quantification of the amount of CTLV in samples was calculated by generating an absolute standard curve to determine the starting number of copies. More specifically, amplicons for CTLV were obtained for each primer set (i.e. F1, 2, and 3 with R) and individually cloned into plasmids (Eurofins MWG Operon, Huntsville, Alabama, USA) (Table 3). The extracted plasmid DNA was linearized using *Hind*III enzyme, to increase the efficiency of dilutions. Serial 10-fold dilution of plasmids carrying a known copy number of CTLV inserts were made to construct a DNA standard curve. The standard curves for CTLV were run in singleplex RT-qPCR setting utilizing 6-carboxyfluorescein FAM fluorophores. Reactions were performed in triplicate to establish the linear response between the Cq values and the log of known copy numbers. The copy numbers for each sample were calculated as described [46]. The slope of the standard curve and the coefficient of determination ($R^2$) were calculated using linear regression [47]. Amplification efficiency (E) was calculated with the formula $E = 10^{(-1/slope)} - 1$ [48, 49]. The intra-assay

variation and inter-assay variations were calculated, by determining the percentage of coefficient of variation (CV %), which was calculated for each sample as follows: mean of the standard deviations of the duplicates divided by the grand mean of the duplicates × 100.

## Comparison of CTLV RT-qPCR detection assay with previously published assays

The newly developed CTLV detection assay was compared to two recently the published RT-qPCR assays. Twenty-two samples from different CTLV isolates and 25 CTLV known negative samples were tested with the SYBR® Green-based RT-qPCR assay by Liu et al. 2011 [35], and the probe-based RT-qPCR assay by Cowell et al. 2017 [25] following the protocols described in each study. Based on the principal that a well performing diagnostic test correctly identifies the diseased individuals in a population, a series of statistical measurements, as reviewed by Bewick et al. 2004 [50], were used to compare the performance of the three RT-qPCR CTLV detection assays. An assay is performing well when sensitivity (Sn) = true positives / (true positives + false negatives) and specificity (Sp) = true negatives / (true negatives + false positives) approach 100%. High positive likelihood ratio ($LR^+$) = sensitivity / (1-specificity) and low (close to zero) negative likelihood ratio ($LR^-$) = (1-sensitivity) / specificity also indicate a well performing diagnostic test. Finally, Youden's index (J) = sensitivity + specificity− 1, can attain the maximum value of 1, when the diagnostic test is perfect and the minimum value of zero, when the test has no diagnostic value [50].

## Results

### Full-length sequences of 12 CTLV isolates via NGS and RACEs

Full-length viral genome sequences of 12 CTLV isolates were obtained by RNA-Seq and the average total reads generated was 27,158,037 which covered 74% to 100% of the viral genome. The full-length cDNA sequences were deposited in GenBank with accession numbers MH108975-MH108986 (Table 1). Excluding the poly (A) tail, the 12 CTLV complete sequences ranged from 6,494 to 6,497 nucleotides (nt) long. Sequence analysis showed the CTLV genome was similar to other capilloviruses, including ASGV and PBNLSV, with two overlapping open reading frames (ORFs) (Fig 1). ORF1 (37–6,354 nt) encoded a 2,105 amino acids (aa) polypeptide, a putative polyprotein around 242-kDa containing methyltransferase-like, papain-like protease, helicase-like, RdRp-like domains, and a coat protein (CP) region (Fig 1). The CP region encoded a 27-kDa protein which was located at the carboxyl-terminal end of the ORF1 polyprotein (5,641–6,354 nt) and was identified based on sequence identity of ASGV CP deposited in GenBank (NC001749) [51]. Two variable regions previously described in ORF1 were also identified (Fig 1) [1, 2]. ORF2 (4,788–5,750 nt) was nested in ORF1 and encoded a 36-kDa protein which belongs to the 30-kDa cell-to-cell movement protein (MP) superfamily (Fig 1).

### Phylogenetic and genomic identity analysis of CTLV full-length sequences

Using three different methods, phylogenetic trees were generated with the available full-length nucleotide sequences of capilloviruses. All three methods generated similar topologies. The neighbor-joining unrooted tree identified four distinct clusters (I—IV) within two well supported clades (A & B) (bootstrap 99%) (Fig 2). Clusters I and II (bootstrap 100%), in clade A, contained CTLV isolates originating from Japan and China along with ASGV isolates from citrus and non-citrus hosts originated from the same geographic locations (Fig 2 and Table 2). Only one of the 12 CTLV isolates from this study (CTLV-IPPN122) was present in clade A

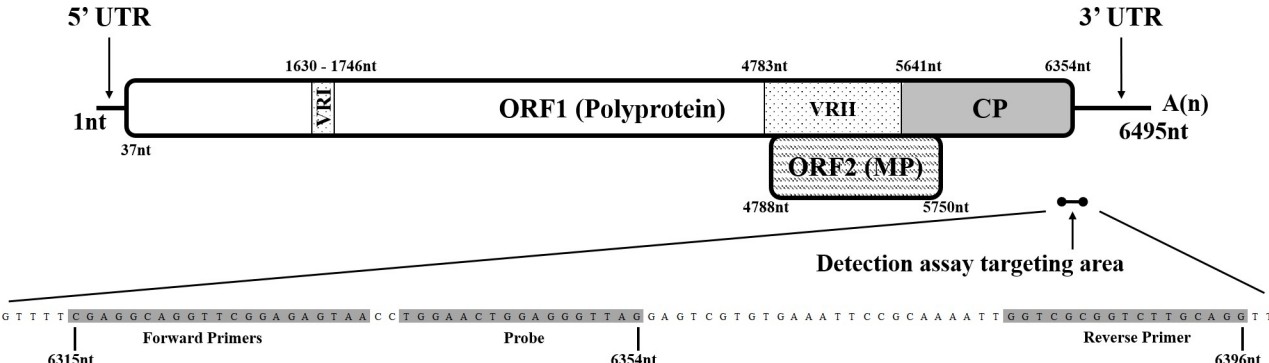

**Fig 1. Schematic representation of the genome organization of citrus tatter leaf virus isolate TL100 (NCBI GenBank Accession No. MH108975).**
Open box represents open reading frame 1 (ORF1) which encoded a 2,105 amino acid (aa) polypeptide, a putative polyprotein around 242-kDa containing methyltransferase-like, papain-like protease, helicase-like, RdRp-like domains, and a coat protein (CP). ORF1 also contains variable region I (VRI) and variable region II (VRII). Open box with backslashes represents open reading frame 2 (ORF2) which is nested in open reading frame 1 and encoded a 36-kDa protein which belongs to 30-kDa superfamily of cell-to-cell movement protein (MP). Solid lines represent the 5' and 3' untranslated regions (UTRs). Short line with end points represent the citrus tatter leaf virus RT-qPCR detection assay targeting region designed in this study.

(cluster I). This isolate was intercepted by the CCPP in a satsuma citrus introduction from China (Fig 2 and Table 2).

The nucleotide sequence identities among the isolates of cluster I ranged within 83.23–93.02% including a 100% identity between ASGV-241KP and ASGV-P-209, both isolated from apple in Japan (Fig 2, Table 2 and Table 4). Sequence identities in cluster II ranged within 94.04–98.47%. Notably, in clade A (clusters I and II), some virus isolates derived from apple (I: ASGV-241KP, and -P-209 and II: ASGV-Li-23), had the highest sequence identities with isolates from lily (II: CTLV-L, 98.47%), citrus (I: CTLV-ASGV-2-HJY, 92.36% and -MTH, 91.07% and II: ASGV-FKSS2, 94.70% and -N297, 94.04%) and citrus relatives (I: ASGV-Nagami, 92.96%) (Fig 2, Table 2 and Table 4). In addition, in cluster I, the isolates ASGV-Nagami from Japan in kumquat (citrus relative, *Fortunella margarita* (Lour.) Swing.) and CTLV-ASGV-2-HJY from China in pummelo (*C. maxima* (Burm.) Merrill) had the highest sequence identity (93.02%) (Fig 2, Table 2 and Table 4).

Clusters III and IV (bootstrap 34%), in clade B, contained 11 of the 12 isolates from this study (Fig 2). In cluster III, three isolates intercepted by the CCPP in citrus introductions from China (i.e. CTLV-TL112, -TL113 and -TL114) grouped with seven CTLV isolates from China and Taiwan, one ASGV citrus isolate from Japan and three ASGV isolates from non-citrus hosts (i.e. apple and actinidia) from China, India and Germany (Fig 2). The nucleotide sequence identities among the isolates of cluster III ranged within 81.49–99.43% including a 100% identity between CTLV-Ponkan8 and CTLV-Pk both isolated from Ponkan mandarin (*C. reticulata* Blanco) in Taiwan (Fig 2, Table 2 and Table 4).

The apple virus isolates in clade B (cluster III) (III: ASGV-AC and ASGVp12) had sequence identities with a virus isolate from actinidia (III: ASGV-Ac) and 22 isolates from citrus and citrus relatives (cluster III and IV) with range of 81.42–82.68% (Fig 2, Table 2 and Table 4). This was in contrast to the high levels of sequence identity observed between apple isolates and lily, citrus and citrus relatives in clade A (91.07–98.47%).

Cluster IV included 11 virus citrus isolates from Japan, China, and the USA. Eight CTLV isolates from this study grouped with two isolates from USA and China and one ASGV citrus isolate from Japan (Fig 2). The nucleotide sequence identities among the isolates of cluster IV ranged within 81.78–99.95% including 100% identity of the CTLV-ML and CTLV-TL111 isolated from Meyer lemon in Florida and CTLV-TL110 isolated from satsuma mandarin

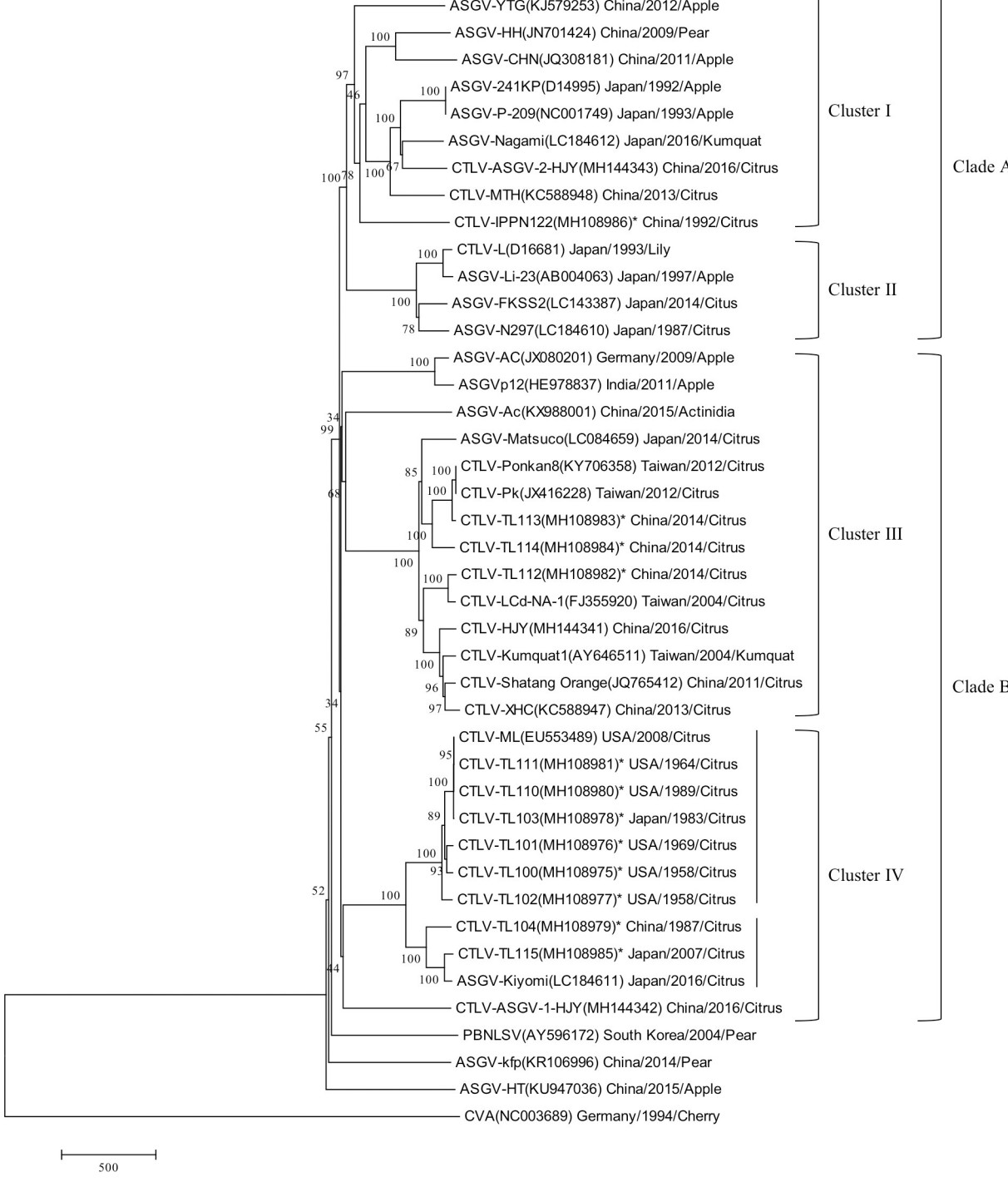

**Fig 2. The unrooted phylogenetic tree based on full-length nucleotide sequences of citrus tatter leaf virus and apple stem grooving virus.**
Total 41 full-length virus genome sequences were used including 12 citrus tatter leaf virus isolates in this study, 12 of citrus tatter leaf virus, 16 isolates of apple stem grooving virus and one isolate of pear black necrotic leaf spot virus from NCBI GenBank database. Cherry virus A was used as outgroup. The tree was constructed by MEGA 7.0.21 using neighbor-joining method with 1000 bootstrap replicates and bootstrap support is indicated at branch points. The scale bar shows the number of substitutions per base. (CTLV: citrus tatter leaf virus; ASGV: apple stem grooving virus; PBNLSV: pear black necrotic leaf spot virus; CVA: cherry virus A).

**Table 4. Full-length nucleotide sequence identities (%) of citrus tatter leaf virus isolates characterized in this study and capilloviruses from NCBI GenBank database.**

| Isolate | Clade | Cluster |
|---|---|---|
| 1-AGSV-YTG | A | I |
| 2-ASGV-HH | | |
| 3-ASGV-CHN | | |
| 4-ASG-24IKP | | |
| 5-ASGV-P-209 | | |
| 6-ASGV-Nagami | | |
| 7-CTLV-ASGV-2-HIY | | |
| 8-CTLV-MTH | | |
| 9-CTLV-PPN122 | | |
| 10-CTLV-L | | II |
| 11-ASGV-Li-23 | | |
| 12-ASGV-FKSS2 | | |
| 13-ASGV-N297 | | |
| 14-ASGV-AC | B | III |
| 15-ASGVp12 | | |
| 16-ASGV-Ac | | |
| 17-ASGV-Matsuco | | |
| 18-CTLV-Ponkan8 | | |
| 19-CTLV-Pk | | |
| 20-CTLV-TL113 | | |
| 21-CTLV-TL114 | | |
| 22-CTLV-TL112 | | |
| 23-CTLV-LGd-NA-1 | | |
| 24-CTLV-HIY | | |
| 25-CTLV-Kumquat1 | | |
| 26-CTLV-Shuang Orange | | |
| 27-CTLV-XBC | | IV |
| 28-CTLV-ML | | |
| 29-CTLV-TL111 | | |
| 30-CTLV-TL110 | | |
| 31-CTLV-TL103 | | |
| 32-CTLV-TL101 | | |
| 33-CTLV-TL100 | | |
| 34-CTLV-TL102 | | |
| 35-CTLV-TL104 | | |
| 36-CTLV-TL115 | | |
| 37-ASGV-Kiyomi | | |
| 38-CTLV-ASGV-1-HIY | | |
| 39-PBNLSV | | Outgroup |
| 40-ASGV-klp | | |
| 41-ASGV-HT | | |
| 42-CVA | | |

*More detailed information can be found in S2 Table.

(*C. unshiu* (Macf.) Marc.) in California. Meanwhile, CTLV-TL103 which was isolated from pummelo in Japan showed 99.95% identity with CTLV-ML, CTLV-TL110, and CTLV-TL111 (Fig 2, Table 2 and Table 4).

Cluster IV contained two subgroups (bootstrap 100%) (Fig 2). The first subgroup contained five CTLV isolates from Meyer Lemon associated with the 1958 introduction of the virus into USA (CTLV-ML, -TL111, -TL101, -TL100 and -TL102). The sequence identities of these isolates ranged within 97.99–98.98% including identical isolates, CTLV-ML and CTLV-TL111, from Florida (Fig 2, Table 2 and Table 4). The California isolates (CTLV-TL101 and -TL102) had 98.56% identity. The isolate from Texas (CTLV-TL100) had 98.52 and 98.98% sequence identity to the isolates from Florida (CTLV-ML and -TL111) and California (CTLV-TL101), respectively (Fig 2, Table 2 and Table 4). The sequence identity of the Meyer Lemon isolates from Florida (CTLV-ML and -TL111) and California (CTLV-TL101 and -TL102) ranged within 97.99–98.70% (Fig 2, Table 2 and Table 4). The second subgroup contained three citrus virus isolates from China (CTLV-TL104) and Japan (CTLV-TL115 and ASGV-Kiyomi) with sequence identities ranged from 95.73 to 98.70% within themselves (Fig 2, Table 2 and Table 4). One China isolate (CTLV-ASGV-1-HJY) stood alone (bootstrap 44%) and had sequence identity of 81.78–82.81% with all other isolates in cluster IV (Fig 2, Table 2 and Table 4).

## Genomic analysis for CTLV RT-qPCR assay design

To analyze the sequence diversity of specific genomic regions, the CTLV genome was divided into three sections: the 5'-UTR and partial polyprotein excluding CP (1–5,640 nt), CP and 3'-UTR (5,641–6,495 nt), and MP (4,788–5,750 nt) (Table 5). The two previously identified variable regions (VRI and VRII) were also analyzed [1, 2].

Sequence identity analysis of the 28 available full genome sequences of the CTLV and ASGV citrus isolates (developed in this study and GenBank) showed that VRI was the most diverse region of the virus genome with 111 variable nucleotide sites among the 117 of the region. In addition, the nucleotide diversity of the VRII was equivalent to that of MP (variable sites 35.08% and 32.81%, respectively) since VRII and MP are essentially covering overlapping areas of the virus genome (Fig 1 and Table 5).

The CP and 3'-UTR (5,641–6,495 nt) was identified as the most conserved region. The percentage of variable nucleotide sites was the lowest (23.63%) and the minimum nucleotide sequence identity was the highest (89.60%) in the virus genome (Table 5). Further analysis revealed that nucleotide sites 6,241–6,440 were the most conserved within the CP and 3'-UTR (Table 6). Therefore, the newly developed RT-qPCR assay was designed to target this 200 nt region (Fig 1, Table 3, and S1 Fig).

## CTLV RT-qPCR assay validation

The applicability, practicability and transferability of this assay was validated by two independent laboratories with consistent reproducible results (Table 7). The assay was also proven to be robust since different annealing temperatures, reaction volumes, qPCR instruments, and master mixes had a minor effect on the Cq values and did not affect the classification of samples as positive or negative (Table 8). The specificity of the assay was determined *in silico* by analyzing the sequence of amplicons from different samples followed by a BLAST search that recognized the amplicon sequences associated only with CTLV. Additionally, the specificity of the assay was evaluated qualitatively with the correct classification (false negative and positive rate 0%) of 112 known CTLV positive and negative samples (Tables 7, 9, 10 and 11). More specifically, the assay detected the virus in 39 known CTLV positive samples from various

**Table 5. Variable sites (%) and nucleotide sequence identities (%) of citrus tatter leaf virus and apple stem grooving virus isolated from citrus and citrus relatives (n = 28).**

| 5'-Untranslated Region and Partial Polyprotein (1–5,640 nt)* | | | Variable Region I (1,630–1,746 nt) | | | Variable Region II (4,783–5,640 nt) | | | Coat Protein and 3'-Untranslated Region (5,641–6,495 nt) | | | Movement Protein (4,788–5,750 nt) | | |
|---|---|---|---|---|---|---|---|---|---|---|---|---|---|---|
| Variable Sites (Variable/Total) | Minimum NSI$ | NSI Mean ± SD | Variable Sites (Variable/Total) | Minimum NSI | NSI Mean ± SD | Variable Sites (Variable/Total) | Minimum NSI | NSH Mean ± SD | Variable Sites (Variable/Total) | Minimum NSI | NSI Mean ± SD | Variable Sites (Variable/Total) | Minimum NSI | NSI Mean ± SD |
| 39.98 (2255/5640) | 79.30 | 84.54 ± 6.66 | 94.87 (111/117) | 34.18 | 54.72 ± 21.80 | 35.08 (301/858) | 82.05 | 87.87 ± 5.25 | 23.63 (202/855) | 89.60 | 92.78 ± 2.92 | 32.81 (316/963) | 83.90 | 88.80 ± 4.77 |

*Nucleotide position is based on reference genome of citrus tatter leaf virus isolate TL100 (NCBI GenBank Accession No. MH108975)

$NSI: Nucleotide Sequence Identity

**Table 6. Variable sites (%) and nucleotide sequence identities (%) of the segmented coat protein and 3'-untranslated region of citrus tatter leaf and apple stem grooving virus isolated from citrus and citrus relatives (n = 28).**

| Position* (nt) | Variable Sites (Variable/Total) | Minimum NSI$ | NSI Mean ± SD |
|---|---|---|---|
| 5641–5840 | 22.50 (45/200) | 86.00 | 94.36 ± 2.71 |
| 5841–6040 | 30.00 (60/200) | 84.00 | 90.03 ± 4.76 |
| 6041–6240 | 25.50 (51/200) | 87.00 | 91.98 ± 3.32 |
| 6241–6440 | 14.50 (29/200) | 92.50 | 95.41 ± 1.99 |
| 6441–6495 | 30.91 (17/55) | 78.18 | 90.49 ± 7.25 |

*Nucleotide position is based on reference genome of citrus tatter leaf virus isolate TL100 (NCBI GenBank Accession No. MH108975)

$NSI: Nucleotide Sequence Identity

geographic locations (Tables 7 and 9) and did not cross-react with 43 known CTLV negative samples of non-inoculated citrus varieties (Table 10) and a series of 30 non-targeted graft-transmissible citrus pathogens (Table 11). When samples were tested with 10-fold serial dilutions (run in triplicate), the sensitivity of the CTLV RT-qPCR showed a linear dynamic range from $10^5$ copies to < 10 copies per μl which indicates the detection assay reached the level of $LOD_6$ with $R^2$ equal to 0.9999 and 100.4% as its efficiency (Fig 3). The mean of viral load was 6.37 x $10^4$ copies of CTLV per μl of infected sample extraction measured by the newly designed CTLV RT-qPCR assay. The CV for CTLV in the RT-qPCR was in the range of 0.23–0.61% (intra-assay variation) and 0.65–1.40% (inter-assay variation) which indicates low variation between different repetitions and different runs.

## Comparison with published CTLV detection assays

The SYBR® Green-based RT-qPCR assay developed by Liu et al. [35] was able to detect CTLV in all 22 samples with the expected melting temperature for the amplicon (81.5–82.0˚C) and its performance measurements (Sn, Sp, $LR^+$, $LR^-$ and J) were optimum and equal to those of the CTLV assay developed in this study (Table 7). The Cq values of the Liu assay were consistently higher than the ones produced from the assay developed in the study (Table 7).

The TaqMan® probe-based RT-qPCR assay designed by Cowell et al. [25] detected CTLV in 15 samples with eight samples having lower Cq values than the assay developed in this study. However, Cowell et al. was unable to detect CTLV in seven samples of three different isolates ($LR^-$ = 0.32) and its performance measurements Sn and J were not optimum (Table 7).

## Discussion

This study presented a systematic approach using the most current technologies for the development and analysis of genomic virus information for the development and validation of a diagnostic assay for CTLV that threatens citrus production worldwide [2, 20, 21].

The data obtained via NGS was *de novo* assembled onto 74% to 100% of the complete CTLV genome which demonstrated the strength of this technology to characterize the virus genome sequence. With RACE sequence data from each isolate, the full-length sequences were assembled in relatively short time compared to traditional sequencing methods. This allowed

**Table 7. Comparison between RT-qPCR assays in detecting citrus tatter leaf virus inoculated and non-inoculated citrus plants.**

| Sample | Experiment | RNA Concentration (ng/µL) | 260 / 280 Ratio | RT-qPCR Cq Value | | | | |
|---|---|---|---|---|---|---|---|---|
| | | | | COX (n = 4) | CTLV This study Lab A[1] (n = 4) | CTLV This study Lab B[2] (n = 2) | CTLV Liu *et al.* 2011 (n = 4) | CTLV Cowell *et al.* 2017 (n = 4) |
| CTLV Isolates (True Positive) | | | | | | | | |
| IPPN122 | TH2986-48 | 165.60 | 1.92 | 12.91 ± 0.04 | 22.16 ± 0.05 | 25.16 ± 0.03 | 27.82 ± 0.26 | 32.54 ± 0.37 |
| TL100 | 1713–1 | 86.24 | 2.24 | 14.83 ± 0.13 | 24.20 ± 0.14 | 26.66 ± 0.58 | 27.11 ± 0.23 | 21.77 ± 0.28 |
| | TL100A | 94.88 | 2.07 | 15.05 ± 0.05 | 24.14 ± 0.06 | 24.92 ± 1.48 | 28.54 ± 0.20 | 22.07 ± 0.11 |
| | TL100B | 46.80 | 2.31 | 15.43 ± 0.06 | 22.55 ± 0.02 | 25.95 ± 1.35 | 26.49 ± 0.23 | 22.55 ± 0.04 |
| TL101 | 1713–2 | 38.72 | 2.60 | 15.98 ± 0.04 | 20.80 ± 0.08 | 23.61 ± 2.39 | 25.24 ± 0.26 | 22.61 ± 0.05 |
| | TL101A | 115.04 | 2.09 | 14.76 ± 0.07 | 21.85 ± 0.10 | 24.09 ± 0.00 | 26.41 ± 0.22 | 21.98 ± 0.08 |
| | TL101B | 41.76 | 2.18 | 15.48 ± 0.26 | 21.62 ± 0.18 | 23.04 ± 1.21 | 25.89 ± 0.24 | 23.00 ± 0.15 |
| | TL101-ND | 130.32 | 2.06 | 14.63 ± 0.22 | 21.78 ± 0.13 | 21.27 ± 0.00 | 26.11 ± 0.14 | 20.75 ± 0.09 |
| TL102 | 2-8-92 | 129.76 | 2.18 | 14.77 ± 0.20 | 22.66 ± 0.27 | 25.14 ± 0.21 | 27.26 ± 0.28 | 21.90 ± 0.04 |
| TL103 | 3288–1 | 139.76 | 2.14 | 14.51 ± 0.10 | 24.71 ± 0.10 | 26.25 ± 0.23 | 28.54 ± 0.02 | - |
| | 3288–2 | 161.04 | 2.27 | 14.51 ± 0.39 | 22.40 ± 0.14 | 24.18 ± 0.26 | 26.20 ± 0.48 | - |
| TL104 | 1855–12 | 73.44 | 2.24 | 15.80 ± 0.09 | 25.62 ± 0.11 | 22.92 ± 1.22 | 28.07 ± 0.20 | - |
| | 2881–1 | 104.72 | 2.05 | 15.19 ± 0.17 | 28.94 ± 0.21 | 26.44 ± 0.11 | 32.76 ± 0.40 | - |
| | 1855–13 | 148.40 | 2.15 | 14.25 ± 0.06 | 27.18 ± 0.09 | 24.64 ± 0.38 | 31.91 ± 0.28 | - |
| TL110 | 3288–3 | 88.64 | 2.06 | 15.26 ± 0.12 | 22.90 ± 0.10 | 22.21 ± 0.06 | 31.01 ± 0.20 | 21.97 ± 0.05 |
| | 3288–4 | 120.64 | 2.11 | 14.81 ± 0.09 | 20.97 ± 0.06 | 23.47 ± 0.41 | 26.89 ± 0.10 | 20.35 ± 0.08 |
| TL111 | 3288–6 | 189.60 | 2.03 | 14.01 ± 0.07 | 24.49 ± 0.08 | 28.22 ± 0.84 | 29.87 ± 0.34 | 21.09 ± 0.15 |
| TL112 | 3291–9 | 140.08 | 2.15 | 14.64 ± 0.17 | 22.75 ± 0.10 | 24.53 ± 0.62 | 28.97 ± 0.21 | 22.66 ± 0.07 |
| TL113 | 3291–10 | 119.12 | 2.24 | 14.75 ± 0.11 | 22.42 ± 0.06 | 23.72 ± 0.13 | 27.91 ± 0.16 | 26.34 ± 0.03 |
| TL114 | 3291–11 | 197.28 | 2.11 | 13.88 ± 0.04 | 23.48 ± 0.09 | 24.54 ± 0.06 | 29.30 ± 0.17 | 24.31 ± 0.12 |
| TL115 | 3170–1 | 221.20 | 2.09 | 13.68 ± 0.18 | 23.22 ± 0.10 | 26.12 ± 0.00 | 26.57 ± 0.13 | - |
| | 3170–2 | 176.00 | 2.22 | 14.06 ± 0.09 | 23.08 ± 0.06 | 27.17 ± 0.30 | 27.55 ± 0.09 | - |
| CTLV-Non-inoculated (True Negative) | | | | | | | | |
| Murcott Mandarin | 1005674 | 44.24 | 2.12 | 16.41 ± 0.11 | - | - | - | - |
| Fortune Mandarin | 3014073 | 57.84 | 2.01 | 16.91 ± 0.10 | - | - | - | - |
| Ponkan Mandarin | 1005802 | 28.48 | 2.34 | 17.08 ± 0.11 | - | - | - | - |
| Cleopatra Mandarin | 1005683 | 40.56 | 2.12 | 17.89 ± 0.06 | - | - | - | - |
| Parson Special Mandarin | 3014062 | 35.44 | 2.00 | 16.52 ± 0.07 | - | - | - | - |
| Tango Mandarin | 1005668 | 39.76 | 2.00 | 15.71 ± 0.12 | - | - | - | - |
| Primosole Mandarin | 1005924 | 17.36 | 2.31 | 15.92 ± 0.04 | - | - | - | - |
| Macetera Sweet Orange | 3014130 | 52.88 | 1.99 | 15.65 ± 0.10 | - | - | - | - |
| Pehrson #3 Valencia | 1005873 | 40.48 | 2.07 | 15.33 ± 0.21 | - | - | - | - |
| Pehrson #4 Valencia | 3014051 | 44.96 | 2.10 | 15.98 ± 0.02 | - | - | - | - |
| Rocky Hill Navel | 1005796 | 47.92 | 2.07 | 16.26 ± 0.07 | - | - | - | - |
| Rio Grande Navel | 1005810 | 53.44 | 2.10 | 15.64 ± 0.10 | - | - | - | - |
| Skaggs Bonanza Navel | 1005797 | 49.92 | 2.02 | 16.19 ± 0.10 | - | - | - | - |
| Autumn Gold Navel | 1005884 | 58.96 | 2.21 | 15.91 ± 0.05 | - | - | - | - |
| China S-9 Satsuma | 1005895 | 61.68 | 2.02 | 15.61 ± 0.10 | - | - | - | - |
| China S-18 Satsuma | 3015105 | 29.92 | 2.38 | 17.05 ± 0.04 | - | - | - | - |
| China S-1 Satsuma | 3015102 | 39.52 | 2.01 | 16.61 ± 0.10 | - | - | - | - |
| China S-17 Satsuma | 3014074 | 12.48 | 3.18 | 15.96 ± 0.09 | - | - | - | - |
| Minneola Tangelo | 1005678 | 56.24 | 2.20 | 17.88 ± 0.04 | - | - | - | - |
| Schaub Rough Lemon | 1005710 | 22.32 | 2.23 | 17.06 ± 0.11 | - | - | - | - |
| Marumi Kumquat | 3014132 | 28.40 | 2.40 | 16.51 ± 0.17 | - | - | - | - |
| Australian Finger Lime | 1005608 | 53.28 | 2.32 | 17.05 ± 0.08 | - | - | - | - |
| Eustis Limequat | 1005814 | 28.40 | 2.38 | 16.49 ± 0.02 | - | - | - | - |
| Valentine Pummelo | 3014144 | 47.60 | 2.34 | 16.57 ± 0.09 | - | - | - | - |
| X639 | 3014082 | 33.84 | 2.42 | 18.63 ± 0.09 | - | - | - | - |
| RT-qPCR Controls | | | | | | | | |
| Positive | H11 / UCD* | NT | NT | 13.71 ± 0.08 | 17.96 ± 0.07 | 23.98 ± 0.37* | 24.38 ± 0.32 | 14.95 ± 0.19 |
| No Template | H9 | - | - | - | - | - | - | - |
| Negative | 861-S-1 | NT | NT | 15.45 ± 0.15 | - | - | - | - |

*(Continued)*

**Table 7.** (Continued)

| Sample | Experiment | RNA Concentration (ng/µL) | 260 / 280 Ratio | RT-qPCR Cq Value | | | | |
|---|---|---|---|---|---|---|---|---|
| | | | | COX (n = 4) | CTLV This study Lab A[1] (n = 4) | CTLV This study Lab B[2] (n = 2) | CTLV Liu *et al.* 2011 (n = 4) | CTLV Cowell *et al.* 2017 (n = 4) |
| RT-qPCR Performance | | | | | CTLV This study Lab A[1] | CTLV This study Lab B[2] | CTLV Liu *et al.* 2011 | CTLV Cowell *et al.* 2017 |
| Sn | | | | | 1.00 | 1.00 | 1.00 | 0.68 |
| Sp | | | | | 1.00 | 1.00 | 1.00 | 1.00 |
| LR+ | | | | | UN | UN | UN | UN |
| LR- | | | | | 0.00 | 0.00 | 0.00 | 0.32 |
| J | | | | | 1.00 | 1.00 | 1.00 | 0.68 |

Abbreviations: Cq: quantitative cycle. CTLV: citrus tatter leaf virus. COX: cytochrome oxidase gene of host plants used as positive internal control [27]. NT: not tested. UN: undefined number (denominator equals 0).

[1]Lab A: Citrus Clonal Protection Program, University of California, Riverside, with ThermoFisher Scientific QuantStudio 12K Flex Real-Time PCR System.

[2]Lab B: Real-Time PCR Research & Diagnostic Core Facility, University of California, Davis, with ThermoFisher 7900HT FAST Real-time PCR system.

*Different positive control was used at Lab B.

for a more comprehensive genome analysis of the CTLV not limited by the available sequences of a small number of virus isolates or parts of the virus genome [1, 2].

The full genome sequence analysis of 28 CTLV and ASGV citrus and citrus relative isolates, developed in this study and available in the GenBank, confirmed the previously reported size, structure and variable regions in the virus genome [1, 2]. Data presented in this study also supported the current taxonomic classification of CTLV as a strain of the ASGV in the *Capillovirus* genus of the *Betaflexiviridae* family since the analysis of multiple full genome sequences of CTLV and ASGV did not meet the species demarcation criteria which is less than 72% nucleotide identity or 80% amino acid identity between their CP or polymerase genes (S8 Table and S9 Table) [52].

**Table 8. Citrus tatter leaf virus RT-qPCR assay validated for robustness.**

| Isolates | Experiment | CTLV RT-qPCR Cq Value | | | | |
|---|---|---|---|---|---|---|
| | | Optimum[1] | Annealing Temperature[2] | | Pipetting Errors[2] | |
| | | 58˚C / 12 µL | -2˚C | +2˚C | -2 µL | +2 µL |
| IPPN122 | TH2986-48 | 22.16 ± 0.05 | 29.59 ± 0.72 | 30.18 ± 0.38 | 30.41 ± 0.25 | 31.62 ± 2.04 |
| TL100 | TL100A | 24.14 ± 0.06 | 23.62 ± 0.06 | 24.29 ± 0.38 | 24.57 ± 0.08 | 23.93 ± 0.35 |
| TL101 | TL101A | 21.85 ± 0.10 | 21.14 ± 0.02 | 21.49 ± 0.09 | 21.45 ± 0.13 | 21.12 ± 0.27 |
| TL103 | 3288–1 | 24.71 ± 0.10 | 25.82 ± 0.25 | 25.81 ± 0.07 | 26.32 ± 0.06 | 26.01 ± 0.09 |
| TL112 | 3291–9 | 22.75 ± 0.10 | 23.44 ± 0.13 | 23.46 ± 0.02 | 23.82 ± 0.35 | 24.61 ± 2.18 |
| TL113 | 3291–10 | 22.42 ± 0.06 | 22.28 ± 0.10 | 22.04 ± 0.11 | 22.29 ± 0.11 | 21.47 ± 0.15 |
| TL115 | 3170–1 | 23.22 ± 0.10 | 24.66 ± 0.15 | 25.47 ± 0.25 | 25.19 ± 0.05 | 25.29 ± 0.10 |

Abbreviations: Cq: quantitative cycle.

[1]Optimum setup was using the conditions validated and optimized in this study including volume, primer probe concentrations, annealing temperature, etc. And the reactions were run on ThermoFisher Scientific QuantStudio 12K Flex Real-Time PCR System.

[2]The RT-qPCR reactions were setup with same concentration of primers and probe and using Bio-Rad iTaq™ Universal Probes One-Step Kit per manufacturer's instruction. The reactions were run on Bio-Rad CFX-96 Real-Time PCR Detection System.

**Table 9. Citrus tatter leaf virus RT-qPCR assay testing citrus tatter leaf virus-inoculated controls.**

| Sample | Origin | CTLV RT-qPCR Cq Value |
|---|---|---|
| FL202 PA A 7/27/10 | FL, USA | 16.62 |
| FL202 Volk sub 1 | FL, USA | 23.09 |
| CTLV #1, FL | FL, USA | 23.32 |
| CTLV #2, FL | FL, USA | 24.50 |
| CTLV #3, FL | FL, USA | 23.33 |
| CTLV #4, FL | FL, USA | 24.94 |
| CTLV #5, FL | FL, USA | 32.82 |
| CTLV #6, FL | FL, USA | 21.28 |
| Positive #1 | South Korea | 19.07 |
| Positive #2 | South Korea | 32.04 |
| Positive #3 | South Korea | 37.61 |
| Positive #4 | South Korea | 25.11 |
| H3 | HI, USA | 26.90 |
| H29 | HI, USA | 26.28 |
| Navel NSW Sample 1 | Australia | 20.98* |
| Navel NSW Sample 2 | Australia | 20.25* |
| Beltsville ARS | MD, USA | 17.83* |

Abbreviations: Cq: quantitative cycle.

*RT-qPCR test was performed at Elizabeth Macarthur Agricultural Institute.

The phylogenetic analysis of the 41 ASGV isolates, revealed four interesting evolutionary and distribution patterns for the virus. First, Asia was highlighted as the point of origin of the virus since countries such as China, Taiwan and Japan were represented in multiple clusters of both phylogenetic clades. This finding also indicated that the origin and diversity of CTLV coincided with the origin of the citrus host. Second, the bottleneck event of the introduction of the virus in the USA from the single citrus variety Meyer Lemon was reflected in cluster IV (first subgroup) in clade B and the high sequence identity (98.52–100%) among the isolates from Texas, Florida, and California. Third, high sequence identities among virus isolates from various citrus producing countries around the world demonstrated the impact of the human activities in the distribution of the virus and the importance of clean stock programs such as CCPP [53]. For example, the CTLV-TL115 isolate was intercepted in an illegal citrus introduction in California (second subgroup, cluster IV, clade B) [54, 55] and it was different from the previously identified isolates of the virus in the state. In addition, the CTLV-IPPN122, -104, -112, -113, and -114 isolates were presented in different variety introductions, separated in time (1987 and 2014), from the original Meyer lemon introduction in 1900s and even though they all originated in China, these isolates clustered in three different phylogenetic clusters (I, III, and IV) in agreement with the principal of high diversity in virus sequences at the point of origin [56–58]. Last but not least, two ASGV spillover events were captured in clade A where ASGV isolates from apple had the highest sequence similarities (91.07–98.47%) with virus isolates from lily, citrus and citrus relatives [59–63]. No spillover event was captured in clade B since sequence identities of apple isolates with actinidia, citrus and citrus relatives was low (81.42–82.68%). Clade B most likely represented the establishment of ASGV in citrus and citrus relatives after its spillover from other species. The spillover events presented here provided some insight to the CTLV ancestry questions for citrus, kumquat, lily and apple presented by Hilf 2008 [32].

**Table 10. Citrus tatter leaf virus RT-qPCR assay testing non-inoculated citrus controls.**

| Citrus Host | Source / Registration number | RT-qPCR Cq Value | |
|---|---|---|---|
| | | COX | CTLV |
| Mandarin (*C. reticulata* Blanco) | | | |
| Murcott Mandarin | 1005674 | 16.49 | - |
| Fortune Mandarin | 3014073 | 17.21 | - |
| Ponkan Mandarin | 1005802 | 16.13 | - |
| Cleopatra Mandarin | 1005683 | 16.56 | - |
| Parson Special Mandarin | 3014062 | 16.24 | - |
| Tango Mandarin | 1005668 | 16.46 | - |
| Primosole Mandarin | 1005926 | 16.99 | - |
| Imperial Mandarin | 3014131 | 16.00 | - |
| Hansen Mandarin | 3014136 | 15.93 | - |
| Encore Ls Mandarin | 3003020 | 15.94 | - |
| Sweet Orange (*C. sinensis* L. Osb.) | | | |
| Macetera Sweet Orange | 3014130 | 16.11 | - |
| Pehrson #3 Valencia | 1005873 | 15.69 | - |
| Pehrson #4 Valencia | 3014051 | 16.58 | - |
| Rocky Hill Navel | 1005796 | 16.50 | - |
| Gillette Navel | 3014134 | 15.55 | - |
| Rio Grande Navel | 1005810 | 17.49 | - |
| Cogan Navel | 1005936 | 16.05 | - |
| Ricalate Navel | 3014068 | 16.93 | - |
| Johnson Navel | 3014096 | 16.47 | - |
| Skaggs Bonanza Navel | 1005797 | 16.93 | - |
| Autumn Gold Navel | 1005884 | 16.42 | - |
| Robertson Navel | 3014125 | 16.51 | - |
| Ceridwen Navel | 3014140 | 16.96 | - |
| Satsuma (*C. unshiu* (Macf.) Marc.) | | | |
| China S-9 Satsuma | 1005895 | 17.39 | - |
| China S-18 Satsuma | 3015105 | 16.11 | - |
| China S-1 Satsuma | 3015102 | 16.52 | - |
| China S-17 Satsuma | 3014074 | 15.85 | - |
| China S-20 Satsuma | 3014064 | 15.95 | - |
| China 6–18 Satsuma | 3014065 | 16.69 | - |
| Tangelo (*C. reticulata* x *C. paradisi*) | | | |
| Minneola Tangelo | 1005678 | 17.05 | - |
| Lemon (*C. limon* (L.) Burm.f.) | | | |
| Schaub Rough Lemon | 1005710 | 16.95 | - |
| Kumquat (*Fortunella* sp.) | | | |
| Centennial Variegated Kumquat | 1005684 | 16.69 | - |
| Nagami Kumquat | 3014145 | 17.40 | - |
| Marumi Kumquat | 3014132 | 16.29 | - |
| Clementine (*C. clementina* Hort. ex Tan.) | | | |
| Fina Sodea Clementine | 3003054 | 16.43 | - |
| Marisol Clementine | 3014101 | 16.72 | - |
| Lime (*C. aurantifolia* (Christm.) Swing.) | | | |
| Australian Finger Lime | 1005608 | 16.95 | - |
| Persian Lime | 1005617 | 15.80 | - |

(*Continued*)

**Table 10.** (Continued)

| Citrus Host | Source / Registration number | RT-qPCR Cq Value | |
| --- | --- | --- | --- |
| | | COX | CTLV |
| Limequat (*Fortunella* sp. x *C. aurantifolia*) | | | |
| Eustis Limequat | 1005814 | 16.66 | - |
| Pummelo (*C. maxima* (Burm.) Merrill) | | | |
| Valentine Pummelo | 3014144 | 16.73 | - |
| Citrange (*P. trifoliata* x *C. sinensis*) | | | |
| Furr C-57 Citrange | 1005930 | 17.51 | - |
| Citron (*C. medica* L.) | | | |
| 'Etrog' Citron Arizona 861-S-1 | 1005966 | 14.02 | - |
| Others | | | |
| X639 | 3014082 | 16.25 | - |

Abbreviations: Cq: quantitative cycle.

Since the genetic variation within the targeted virus population can lead to false negative RT-qPCR results, for the design of the CTLV detection assay we aimed to locate the most conserved region on the virus genome beyond the traditional approaches that focus on individual genes presumed conserved due to their function [64]. The newly developed detection assay was further validated according to the guidelines for validation of qualitative real-time PCR methods and its performance was assessed with statistical measurements [50, 65]. We showed that the most conserved CTLV genome region was not confined in a single gene, but it spanned the region between the CP gene and 3'-UTR, thus it was targeted for the RT-qPCR assay design. The conserved nature of the CTLV CP could be a result of its function in virion assembly [64]. And for the 3'-UTR of CTLV, the high identity among isolates indicates that it has an important role in CTLV replication and/or translation [66].

Compared to published CTLV qPCR assays that were designed on limited or single isolate sequences, the assay in this study performed better (e.g. Youden's index) and detected a diverse range of CTLV isolates from different geographic locations, citrus varieties, and isolation times, because it was designed using a high number of virus sequences [25, 34, 35]. These results agree with Roussel et al. [67] who reported, that the RT-qPCR designed for prune dwarf virus (PDV) failed to detect many virus isolates because the assay was designed from very few published PDV sequences in the GenBank. In addition, the sensitivity and specificity of this assay was improved by using MGB probes [68, 69], designed from the multiple sequence alignment, that targeted the identified conserved genomic region between the CP gene and 3'-UTR. Furthermore, measuring the intra and inter assay variations confirmed the reproducibility and repeatability of the developed RT-qPCR assay. Finally, measuring viral loads and performing reactions under variable conditions showed that the newly developed RT-qPCR is robust and can detect minimal quantities of the CTLV.

Next generation sequencing (NGS) technologies combined with bioinformatics analysis have proven to be powerful tools in identifying and characterizing novel sequences of pathogens, in studying disease occurrence, genome variability, and phylogeny [38–40]. Using NGS technologies within a well-defined qPCR design, development and validation protocol [41, 42] is that qPCR assays can be regularly updated as more target pathogen genomes are sequenced, therefore, increasing the value of the assay in preventing virus outbreaks and managing virus spread and induced disease.

**Table 11. Citrus tatter leaf virus RT-qPCR assay testing samples inoculated with non-targeted citrus pathogens.**

| Citrus Pathogen Isolate | Source / Registration number | RT-qPCR Cq Value | | |
|---|---|---|---|---|
| | | COX | CTLV | Target |
| Citrus tristeza virus (CTV) | | | | |
| T514 | T514-2 | 14.55 | - | 25.75 |
| T538 | 3275–4 | 14.23 | - | 22.46 |
| SY568 | 2761–114 | 13.87 | - | 20.30 |
| Citrus psorosis virus (CPsV) | | | | |
| P201 | 1766–5 | 14.56 | - | 26.52 |
| P203 | 2-26-98 | 14.93 | - | 28.38 |
| P218 | 3175–2 | 14.06 | - | 28.67 |
| Citrus leaf blotch virus (CLBV) | | | | |
| CLBV, Spain | 3069–1 | 14.86 | - | 27.89 |
| Citrus vein enation virus (CVEV) | | | | |
| VE702 | 2923–2 | 14.46 | - | (+)* |
| VE703 | 2923–3 | 14.33 | - | (+)* |
| VE704 | 2923–4 | 14.42 | - | (+)*x |
| Citrus yellow vein virus (CYVV) | | | | |
| YV3163-1 | 3163–1 | 14.54 | - | 21.28 |
| YV3163-3 | 3163–3 | 15.09 | - | 20.72 |
| YV920C | 3163–20 | 14.66 | - | 21.94 |
| Infectious variegation virus (IVV) | | | | |
| IV400 | IV400 3-26-03 | 13.82 | - | 13.63 |
| IV401 | IV401A 1993 | 14.62 | - | 28.41 |
| Concave gum | | | | |
| CG302 | CG302 7-8-04 | 14.30 | - | (+)** |
| CG308 | 2355–4 | 14.58 | - | (+)** |
| CG309 | CG309 11-14-96 | 13.86 | - | (+)** |
| Citrus viroids | | | | |
| Citrus exocortis viroid (CEVd) | 2765–1 | 14.43 | - | 26.17 |
| Citrus bent leaf viroid | 2765–2 | 14.69 | - | 26.46 |
| Citrus bent leaf viroid- LSS | 3237–3 | 17.05 | - | 32.76 |
| Hop stunt viroid, non-cachaxia | 2765–4 | 14.92 | - | 21.62 |
| Hop stunt viroid, cachaxia | 2765–6 | 17.39 | - | 27.34 |
| Citrus dwarfing viroid | 2765–12 | 14.88 | - | 27.92 |
| Citrus bark cracking viroid | 3200–1 | 15.13 | - | 23.45 |
| Citrus viroid V | 3195–5 | 13.54 | - | 26.32 |
| *Candidatus* Liberibacter | | | | |
| asiaticus | HLB B | 17.49 | - | 26.69 |
| asiaticus | HLB G | 16.78 | - | 29.55 |
| *Spiroplasma citri* | | | | |
| C189 | C189 7-8-09 | 16.95 | - | 30.11 |
| S616 | S600 7-8-09 | 17.09 | - | 29.26 |

Abbreviations: Cq: quantitative cycle.

*Citrus vein enation virus was tested by conventional RT-PCR

**Concave gum was tested by biological indexing

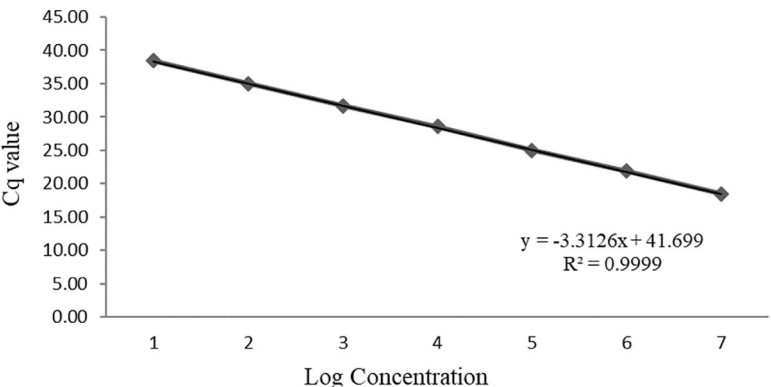

**Fig 3. Standard curve analysis of RT-qPCR sensitivity.** The X-axis displays the log concentration and the Y-axis represents the value of quantitative cycle (Cq).

We propose that in the era of powerful affordable sequencing platforms the presented approach of full-genome sequence analysis of multiple virus isolates, and not only a small genome region of a small number of virus sequences, becomes a guideline for the design and comprehensive validation of qPCR-based virus detection assays especially for use in high value germplasm programs [26, 30, 31]. We understand the academic urgency for scientific publications however specifically in the case of diagnostics that affect international trade, quarantines and regulatory decisions that by extension affect the livelihoods of thousands of people, we urge the research community to dedicate the necessary resources and time for the appropriate design and validation of pathogen detection assays. We hope that this publication offers a valuable case study for such consideration.

## Supporting information

**S1 Fig. Citrus tatter leaf virus detection assay targeting region.** Multiple nucleotide sequences alignment of citrus tatter leaf virus and apple stem grooving virus isolated from citrus and citrus relatives host. Citrus tatter leaf virus detection assay targeting region (highlighted in dark grey) and primers-probe set are also shown. Apple stem grooving virus isolate P-209 is used here to represent the species.
(PDF)

**S1 Table. Oligonucleotide primers used in this study.**
(PDF)

**S2 Table. Full-length nucleotide sequence identities (%) of citrus tatter leaf virus isolates in this study and capilloviruses from NCBI GenBank database.**
(PDF)

**S3 Table. Nucleotide sequence identities (%) of 5'-untranslated region (5'-UTR) and polyprotein (not including coat protein region).**
(PDF)

**S4 Table. Nucleotide sequence identities (%) of coat protein (CP) and 3'-untranslated region (3'-UTR).**
(PDF)

**S5 Table. Nucleotide sequence identities (%) of movement protein (MP).**
(PDF)

**S6 Table. Nucleotide (below diagonal) and amino acid (above diagonal) sequences identities (%) of variable region I (VRI) of citrus tatter leaf virus and apple stem grooving virus isolated from citrus and citrus relatives.**
(PDF)

**S7 Table. Nucleotide (below diagonal) and amino acid (above diagonal) sequences identities (%) of variable region II (VRII) of citrus tatter leaf virus and apple stem grooving virus isolated from citrus and citrus relatives.**
(PDF)

**S8 Table. Nucleotide (below diagonal) and amino acid (above diagonal) sequences identities (%) of polyprotein (PP).**
(PDF)

**S9 Table. Nucleotide (below diagonal) and amino acid (above diagonal) sequences identities (%) of coat protein (CP).**
(PDF)

**S10 Table. Nucleotide (below diagonal) and amino acid (above diagonal) sequences identities (%) of movement protein (MP).**
(PDF)

## Acknowledgments

The authors are grateful to all past and current CCPP personnel for their dedicated work and especially for creating and maintaining the *in planta* CTLV collection through time. We acknowledge Samantha Mapes at the Real-time PCR Research and Diagnostic Core Facility, UC Davis for her excellent technical assistance. We also would like to thank Dr. Robert Krueger and Dr. MaryLou Polek from the USDA National Clonal Germplasm Repository for Citrus for sharing virus isolates as well as Dr. William Dawson (Citrus Research and Education Center, University of Florida, U.S.A), Dr. Nerida Donovan (Elizabeth Macarthur Agricultural Institute, Australia), Dr. Jae Wook Hyun (Citrus Research Institute, Korea), Dr. Michael Melzer (Department of Plant & Environmental Protection Sciences, University of Hawaii, USA), and Benjamin Rosson (Bureau of Citrus Budwood Registration, Florida Department of Agriculture and Consumer Services, USA) who provided virus isolates for assay validation.

## Author Contributions

**Conceptualization:** Shih-hua Tan, Fatima Osman, Sohrab Bodaghi, Tyler Dang, Greg Greer, Georgios Vidalakis.

**Data curation:** Shih-hua Tan, Fatima Osman, Sohrab Bodaghi, Tyler Dang, Greg Greer, Amy Huang, Sarah Hammado, Shurooq Abu-Hajar.

**Formal analysis:** Shih-hua Tan, Fatima Osman, Tyler Dang.

**Funding acquisition:** Georgios Vidalakis.

**Investigation:** Georgios Vidalakis.

**Methodology:** Shih-hua Tan, Fatima Osman, Tyler Dang, Georgios Vidalakis.

**Project administration:** Georgios Vidalakis.

**Resources:** Greg Greer, Georgios Vidalakis.

**Software:** Tyler Dang.

**Supervision:** Shih-hua Tan, Fatima Osman, Sohrab Bodaghi, Georgios Vidalakis.

**Validation:** Shih-hua Tan, Fatima Osman, Sohrab Bodaghi, Amy Huang, Sarah Hammado, Shurooq Abu-Hajar, Georgios Vidalakis.

**Visualization:** Shih-hua Tan, Fatima Osman.

**Writing – original draft:** Shih-hua Tan, Fatima Osman.

**Writing – review & editing:** Shih-hua Tan, Fatima Osman, Sohrab Bodaghi, Tyler Dang, Greg Greer, Amy Huang, Sarah Hammado, Shurooq Abu-Hajar, Roya Campos, Georgios Vidalakis.

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
