## [Decision Letter · Decision Letter 0]

19 Aug 2019

PONE-D-19-16252

Full genome characterization of 12 citrus tatter leaf virus isolates for the development of a detection assay

PLOS ONE

Dear Ms. Tan,

Thank you for submitting your manuscript to PLOS ONE. After careful consideration, we feel that it has merit but does not fully meet PLOS ONE’s publication criteria as it currently stands. Therefore, we invite you to submit a revised version of the manuscript that addresses the points raised during the review process.

We would appreciate receiving your revised manuscript by Oct 01 2019 11:59PM. To enhance the reproducibility of your results, we recommend that if applicable you deposit your laboratory protocols in protocols.io, where a protocol can be assigned its own identifier (DOI) such that it can be cited independently in the future. For instructions see: http://journals.plos.org/plosone/s/submission-guidelines#loc-laboratory-protocols

We look forward to receiving your revised manuscript.

Kind regards,

Ulrich Melcher

Academic Editor

PLOS ONE

Journal Requirements:

Additional Editor Comments (if provided):

I was only able to obtain an evaluation of this manuscript from one reviewer*busy time of year). Consequently, I read the submission carefully myself. The manuscript describes an excellent description of analysis of viral genomic datasets. I did find a few places for which I think changes are needed. I am assuming the authors will make these small changes and those suggested by Reviewer1 on the way to production.

Missing spaces:

l. 33 Analysis/highlighted

l. 125 the/3’ end

Wherever quantities of units are displayed, a space is usually required between the quantity and the unit.

I object to use of the term % homology. A pair of sequences are homologous or they are not. There is no in between. They may be a certain percentage identical or % simoilar.

Reviewers' comments:

Reviewer's Responses to Questions

**Comments to the Author**

1. Is the manuscript technically sound, and do the data support the conclusions?

Reviewer #1: Yes

2. Has the statistical analysis been performed appropriately and rigorously? 

Reviewer #1: Yes

3. Have the authors made all data underlying the findings in their manuscript fully available?

Reviewer #1: Yes

4. Is the manuscript presented in an intelligible fashion and written in standard English?

Reviewer #1: Yes

5. Review Comments to the Author

Reviewer #1: General Comments:

The manuscript titled “Full genome characterization of 12 citrus tatter leaf virus isolates for the development of a detection assay” presents data on full-length genome sequences of 12 CTLV isolates from different geographical areas, intercepted and maintained for the past 60 years at the Citrus Clonal Protection Program (CCPP), University of California, Riverside. The manuscript is well written and provides useful information and a good reference point for future works regarding design and validation of plant virus detection assays.

Specific comments: minor corrections needed.

Page 9, Line 165: introduce comma (,) after “citrus tissues” and before “an RT-qPCR”

Page 10, Line 190: replace “annealing cycle” with “amplification cycle”

Page 12, Line 232: start the sentence with “Twenty-two” instead of “22”

Page 21, Table 7: Cq Values of COX are consistently different between true positive samples to true negative samples. Though, the COX results presented here does not have any significant bearing on this data analysis but out of curiosity, I would like to know any explanation for the different Cq values observed.

Page 27, Line 441-442: Authors talk about Liu assay. However, I do not see the Cq values of Liu assay. It is possible that the Table 7 is incomplete or a column (for Liu assay) is missing in the reviewer’s copy.

Page 21, Table 7: Similarly, Lab B data is not displayed in the Reviewer’s copy or it is missing from the table.

Page 33, Line 469: Since the center of origin for citrus is Asia, it is not only surprising to find diverse citrus cultivars but also high genetic diversity of CTLV from Asia. Authors may want to add a sentence or so to reflect the point.

6. PLOS authors have the option to publish the peer review history of their article (what does this mean?). If published, this will include your full peer review and any attached files.

Reviewer #1: No

---

## [Author Response · Author response to Decision Letter 0]

6 Sep 2019

Dear Editor and Reviwer,

We wish to submit the revised manuscript, PONE-D-19-16252 “Full genome characterization of 12 citrus tatter leaf virus isolates for the development of a detection assay” by Tan et al. to the PLOS ONE.

We took in consideration all the comments from you and the Reviewer and made the changes accordingly in the manuscript and summarized in the attached response to your review. Due to the PLOS ONE table format requirements, Table 7 and some of the other tables may not be viewed properly in “Print Layout” mode of the word file. Please select “Draft” mode under “View” section in the Microsoft Word to view tables properly. We also attached Table 7 in the last page of the cover letter for your and the Reviewer’s convenience.

We would like to thank you and the Reviewer again for your excellent recommendations and we trust that the revisions made to the manuscript are sufficient to warrant publication.

Best Regards,

Shih-hua Tan & Georgios Vidalakis

Department of Microbiology and Plant Pathology

University of California, Riverside

Note: For detailed info, please find the pdf file with title "Rebuttal Letter with Response to Reviewers" in the re-submission package. Thank you.

--

Response to Reviewers

Journal Requirements:

Additional Editor Comments (if provided):

I was only able to obtain an evaluation of this manuscript from one reviewer*busy time of year). Consequently, I read the submission carefully myself. The manuscript describes an excellent description of analysis of viral genomic datasets. I did find a few places for which I think changes are needed. I am assuming the authors will make these small changes and those suggested by Reviewer1 on the way to production.

Missing spaces:

l. 33 Analysis/highlighted

l. 125 the/3’ end

Wherever quantities of units are displayed, a space is usually required between the quantity and the unit.

I object to use of the term % homology. A pair of sequences are homologous or they are not. There is no in between. They may be a certain percentage identical or % similar.

Authors: 

We thank the Editor for his time and effort in reviewing this manuscript during this busy time of the year as well as for providing useful comments. We have made the recommended changes and used the “sequence identity analysis” with % identical instead of homology analysis or homologous throughout the revised manuscript indicated with track changes.

Reviewers' comments: (Question 5. Review Comments to the Author)

Reviewer #1: General Comments:

The manuscript titled “Full genome characterization of 12 citrus tatter leaf virus isolates for the development of a detection assay” presents data on full-length genome sequences of 12 CTLV isolates from different geographical areas, intercepted and maintained for the past 60 years at the Citrus Clonal Protection Program (CCPP), University of California, Riverside. The manuscript is well written and provides useful information and a good reference point for future works regarding design and validation of plant virus detection assays.

Specific comments: minor corrections needed.

• Page 9, Line 165: introduce comma (,) after “citrus tissues” and before “an RT-qPCR”

• Page 10, Line 190: replace “annealing cycle” with “amplification cycle”

• Page 12, Line 232: start the sentence with “Twenty-two” instead of “22”

Authors: 

We thank the reviewer for his comments and recommended corrections. We have made the recommended changes in the revised manuscript.

Note: For detailed info, please find the pdf file of "Rebuttal Letter with Response to Reviewers" in the re-submission package. Thank you.

• Page 21, Table 7: Cq Values of COX are consistently different between true positive samples to true negative samples. Though, the COX results presented here does not have any significant bearing on this data analysis but out of curiosity, I would like to know any explanation for the different Cq values observed.

Authors: 

This is an interesting observation. We thank the reviewer for his comment. We agree that COX Cq values have no significant bearing on the data analysis of true positive and true negative. We are using the COX assay as designed by Osman et al. (2015) based on the housekeeping gene, cytochrome oxidase (COX; GenBank Accession No. CX297817), to assess the integrity and quality of the nucleic acids after the CCPP developed semi-automatic nucleic acid extraction procedure for citrus tissue. From our experience, testing hundreds of samples from different citrus species, COX Cq values between 12 and 24 give reliable PCR results for citrus virus detection as we experienced with CTLV in this study and other viruses in previous work (Osman et al. 2015). In addition, we have observed that similar range of Cq values for other citrus housekeeping genes (e.g. NADH) provided reliable citrus viroids detection when testing tens of thousands of samples in our lab for the California Department of Food and Agriculture (CDFA) Citrus Nursery Stock Pest Cleanliness Program (CDFA data). We hope we provided some good information to the Reviewer. If you need any additional information, please let us know.

• Page 27, Line 441-442: Authors talk about Liu assay. However, I do not see the Cq values of Liu assay. It is possible that the Table 7 is incomplete or a column (for Liu assay) is missing in the reviewer’s copy.

• Page 21, Table 7: Similarly, Lab B data is not displayed in the Reviewer’s copy or it is missing from the table.

Authors: 

We apologize to the Reviewer for the inadequate presentation of Table 7. Unfortunately, this was the PLOS ONE format issue. In some cases, tables cannot be viewed properly in “Print Layout” mode of Word file. Please select “Draft” mode under “View” section in Microsoft Word to see the full table. We are also attaching Table 7 at the end of this response for your convenience.

• Page 33, Line 469: Since the center of origin for citrus is Asia, it is not only surprising to find diverse citrus cultivars but also high genetic diversity of CTLV from Asia. Authors may want to add a sentence or so to reflect the point.

Authors: 

We thank the Reviewer for his recommendation. We have made the recommended addition in the revised manuscript: page 28 line 638 “This finding also indicated that the origin and diversity of CTLV coincided with the origin of the citrus host”.

Note: For detailed info, please find the pdf file with title "Rebuttal Letter with Response to Reviewers" in the re-submission package. Thank you.

---

## [Editor Report · Decision Letter 1]

3 Oct 2019

Full genome characterization of 12 citrus tatter leaf virus isolates for the development of a detection assay

PONE-D-19-16252R1

Dear Dr. Tan,

We are pleased to inform you that your manuscript has been judged scientifically suitable for publication and will be formally accepted for publication once it complies with all outstanding technical requirements.

With kind regards,

Ulrich Melcher

Academic Editor

PLOS ONE
---

## [Editor Report · Acceptance letter]

8 Oct 2019

PONE-D-19-16252R1 

Full genome characterization of 12 citrus tatter leaf virus isolates for the development of a detection assay 

Dear Dr. Tan:

I am pleased to inform you that your manuscript has been deemed suitable for publication in PLOS ONE. Congratulations! Your manuscript is now with our production department. 

With kind regards,

on behalf of

Dr. Ulrich Melcher 

Academic Editor

PLOS ONE